# Added Alternatives in Spoken Interaction: A Corpus Study on German *Auch*

**Laura Reimer ***  **and Christine Dimroth**

Institute of German Studies, University of Münster, 48143 Münster, Germany; christine.dimroth@uni-muenster.de
\* Correspondence: laura.reimer@uni-muenster.de

**Abstract:** Particles such as German *auch* ('also') establish an additive relation between expressions in their scope (added constituent, AC) and context alternatives against the background of shared information (common denominator). In spoken interaction, however, explicit alternatives are not necessarily present and expressions can be construed as alternatives against different variants of a common denominator. It is the aim of the present paper to investigate to what extent the presence of alternatives influences the construction of utterances containing an additive particle. This is particularly relevant for German, where speakers can choose between an unstressed and stressed version of *auch*. We ask whether properties of the alternatives and their common denominators influence the choice to use stressed or unstressed *auch*. In a corpus study on spoken language, we classified the versions of *auch*, the particles AC, the alternatives in the preceding context and their common denominator. The results show that the speaker's choice is influenced by the relation of the utterance to context alternatives. Specifically, the degree of explicitness of alternatives, the number of alternatives, and the degree of abstractness of the common denominator influence the continuation of the discourse, measured by the preference for one of the two variants of the particle *auch*.

**Keywords:** alternatives; additive particles; corpus study; discourse; context

## 1. Introduction

Additive particles such as the English *too*, *also*, or German *auch* (Krifka 1999; Reis and Rosengren 1997; Dimroth 2004; Sæbø 2004; Sudhoff 2010) establish an additive relation between expressions in their scope (i.e., the added constituent, AC) and comparable context expressions (i.e., the alternatives) against the background of shared information (the common denominator). Alternatives do not exist in the void—they are always alternatives with respect to something. In textbook examples, the sentence containing the additive particle and the context sentence typically share an explicit common denominator, that is, they contain identical material (underlined) and differ only with respect to the particle's AC (in brackets) and its context alternative (italic); cf. (1) from Krifka (1999):

1.   *Peter* <u>ate pasta</u> and [Pia] <u>ate pasta</u>, **too**.

In (1), situations with the descriptive properties 'ate pasta' are the common denominator for the AC *Pia* and the alternative *Peter*. In spoken interaction, however, explicit alternatives are not necessarily present and expressions can be construed as alternatives against different variants of the common denominator. As a consequence, identifying relevant alternative(s) can be challenging. It is the aim of the present paper to identify alternatives in spoken interaction, and to investigate to what extent the presence of alternatives influences the construction of an utterance containing an additive particle. This is particularly relevant for German, where speakers can choose between an unstressed (see 2) and stressed (see 3) version of the additive particle *auch*.

2.   Peter hat Pasta gegessen. Ich frage mich, ob **auch** [Pia] Pasta gegessen hat.
     'Peter ate pasta. I am wondering whether Pia ate pasta, too.'

3.   Peter hat Pasta gegessen. Ich frage mich, ob [Pia] **AUCH** Pasta gegessen hat.
     'Peter ate pasta. I am wondering whether Pia ate pasta, too.'

In this paper, we ask whether properties of the alternatives and their common denominators influence the choice to use stressed or unstressed *auch*. To that end, we conducted a corpus study on spoken language where we classified the versions of *auch*, the particles' AC, the alternatives in the preceding context and their common denominator. The results show that the speaker's choice is influenced by the relation of the utterance to context alternatives. Specifically, the degree of explicitness of alternatives, the number of alternatives, and the degree of abstractness of the common denominator influence the continuation of the discourse, measured by the choice of the two variants of the particle *auch*. The paper is structured as follows: In Section 2, we offer some remarks about the distinction between unstressed and stressed *auch* with respect to discourse alternatives and the common denominator, and about the distinction between the focus particle reading and another frequent reading of the lexical element *auch*. We will also present previous corpus studies on focus particles and focus alternatives. In Section 3, we will turn to the methodology of our corpus study and Section 4 reports on the findings of the corpus study. Finally, Section 5 discusses the results.

## 2. Background

### 2.1. Unstressed and Stressed Auch

As stated above, the focus particle *auch* associates with a constituent (the added constituent, AC), which is related to one or more alternative expression(s)[1] in the context (Krifka 1999; Reis and Rosengren 1997; Dimroth 2004; Sæbø 2004; Sudhoff 2010). The sentence containing the focus particle *auch* and the context sentence containing the discourse alternative share some information, which we label the common denominator. In text book examples, as the one in (1), the common denominator can be an exact repetition, that is, the information of the sentence containing the focus particle and the AC is literally repeated (see Schwarzschild 1999). Corpus data show, however, that there is not always an exact repetition, as in (1). Sometimes, it is sufficient that a more abstract frame of comparison can be established in some way. For example, an additive particle can also be used when the property of the alternative ('ate pasta') that was asserted for the alternative is merely under discussion with respect to the AC as in (4), when it is entailed as in (5), or when it is somehow reinterpreted as in (6) (see Baumann and Riester 2012 for an overview). In (6), it can be inferred that Peter might have been hungry, since in order to eat pasta, one usually has to be hungry (see Prince 1981). In these cases, there are some degrees of freedom concerning the common denominator, indicating that speakers can establish additive relations on the basis of abstract frames of comparison.

4.   Peter hat Pasta gegessen. Ich frage mich, ob **auch** [Pia] Pasta gegessen hat.
     'Peter ate pasta. I am wondering whether Pia ate pasta, too'.
5.   Peter hat Pasta gegessen. Ich frage mich, ob **auch** [Pia] italienisches Essen gegessen hat.
     'Peter ate Pasta. I am wondering whether Pia ate Italian food, too'.
6.   Peter hat Pasta gegessen. Ich frage mich, ob **auch** [Pia] Hunger hatte.
     'Peter ate pasta. I am wondering whether Pia was hungry, too'.

In spoken interaction, there is also flexibility concerning the discourse alternatives, in that explicit alternatives do not necessarily have to be present in the context (see 7).

7.   Ich muss noch mehr Gläser für die Party besorgen. **Auch** meine alten Mitbewohner wollen plötzlich kommen.
     'I have to buy more glasses for the party. My old roommates want to come as well.'

In (7), the AC of *auch* is added to an open set with unspecified members, such as {other people/my new roommates/my old roommates}. The relevant discourse alternatives cannot easily be named or counted. Another example is '*(Hey) X have feelings too(!)*' by Beaver and Zeevat (2007), where *X* can be e.g., *men*, *fish* or *we*. In this example, *too* occurs

without a linguistic antecedent. In this case, the assumption that there is some *Y* (e.g., *women*, *people*, *you*) such that *Y* has a unique salient contrast with *X* and such that *Y has feelings* is already in the common ground, or it is accommodated.

Due to the variability of the common denominator and the context alternatives, identifying the relevant alternative(s) in authentic data can be a challenge for researchers. The interlocutors, however, do seemingly produce and comprehend the particles and the corresponding discourse relations in an effortless way.

It is the aim of the present study to investigate whether the number and explicitness of discourse alternatives influence the construction of utterances containing the German particle *auch*. Speakers can choose between two ways of integrating the particle *auch* relative to its AC. One way is using unstressed *auch* in a position preceding the AC (as in 4–7), and the other way is by using stressed *auch* in a position following the AC, as indicated by capital letters in the remainder of this paper (see 8).

8.    Peter hat Pasta gegessen. Ich frage mich, ob [Pia] **AUCH** Pasta gegessen hat.
      'Peter ate pasta. I am wondering whether Pia ate pasta, too.'

We ask, whether there are differences between stressed and unstressed additive *auch* with respect to the requirement of explicit contextual alternatives. In many cases, unstressed *auch* and stressed *AUCH* are compatible with similar discourse contexts. This becomes obvious when examples (2) and (3) are compared. However, there are also cases where one version is less acceptable or even unacceptable, while the other version is perfectly acceptable, as shown by (9).

9.    a. Es ist anzunehmen, dass der Anstieg der Arbeitslosenzahlen **auch** [die Bundeskanzlerin] beschäftigt.
      b. #Es ist anzunehmen, dass der Anstieg der Arbeitslosenzahlen [die Bundeskanzlerin] **AUCH** beschäftigt.[2]
      'It can be assumed that the increase in unemployment also bothers the chancellor.'

While (9a) is acceptable, (9b) is marked or even unacceptable. A reason could be that the context does not contain an explicit set of alternatives, but that the chancellor is added to an unspecified set of alternatives. This example suggests that stressed *AUCH* is not easily compatible with an alternative set being vague. According to Reis and Rosengren (1997), there are systematic (but subtle) meaning differences between unstressed *auch* and stressed *AUCH*. The meaning of unstressed *auch*, which has new material to its right, corresponds to the utterance meaning FURTHERMORE, whereas stressed *AUCH*, which also scopes to the right and has only given material in its scope, corresponds to the utterance meaning LIKEWISE. In the case of FURTHERMORE, the AC of *auch* is simply added to a set of alternatives, similar to an enumeration (see 10). In the case of LIKEWISE, the AC and the alternative are both sharing a specific property (e.g., ate pasta, see 11). The best fit for the FURTHERMORE reading might be a set of several alternatives, while the best fit for the LIKEWISE reading might be one alternative.

10.   Peter, Paul und John haben Pasta gegessen. **Auch** Pia hat Pasta gegessen.
      'Peter, Paul and John ate pasta. Pia ate pasta, too.'

11.   Peter hat Pasta gegessen. Genau wie Pia, die hat **AUCH** Pasta gegessen.
      'Peter ate Pasta. Just like Pia, she ate pasta, too.'

That a restricted set of alternatives is involved in (11) is compatible with the assumption that the AC of stressed *AUCH* has the information structural status of a (contrastive) topic (see Krifka 1999; Dimroth 2004; Sæbø 2004). This suggests that the relevant alternative to Pia must somehow already be under discussion. The additive particle thus marks that a similar claim is made about two distinct units of topical information. Due to the "under discussion" constraint, there should thus be less alternatives and they should be more explicit than is necessary for unstressed *auch*. We ask whether properties of (a) the alternatives and (b) their common denominators influence the choice of the speaker to use one version of the particle *auch* over the other. This is particularly interesting since focus

alternatives play a vital role in language comprehension (e.g., Gotzner et al. 2013; Gotzner et al. 2016; Spalek et al. 2014) and language production (e.g., Kaiser 2010).

In German, there is an additional reading of *auch* that does not have an additive meaning, namely that of the modal particle[3] (see 12).

12.   Pia ist nicht mehr hungrig, aber sie hat **auch** Pasta gegessen.
      'Pia is not hungry anymore, but this is not surprising, since she ate pasta.'

Since we are interested in the focus particle meaning of *auch*, we have to target the modal particle meaning as well. The reason is the following: German particles are highly ambiguous and former studies on German additive particles do not take modal particle readings into account. We pursue the goal of distinguishing between both meanings in order to only include those cases that have the intended additive meaning. For that reason, the following part of the paper gives some background information on the features of German modal particles, based on the example in (12).

(a) In (12), *auch* does not associate with a constituent, but scopes over the whole proposition *p* (*Pia ate pasta*). Thus, the lack of an AC is one characteristic of the modal particle. (b) Since a potential alternative is always related to an AC, the lack of an AC leads to a lack of an alternative in the context, which is another characteristic of the modal particle. (c) The function of *auch* as a modal particle is to signal that there is no contrast between the context *q* (*Pia is not hungry*), and *p* (*Pia ate pasta*), or that *q* is not surprising in the light of *p* (see Thurmair 1989; Schmitz et al. 2018). Thus, it should always be possible to replace the modal particle *auch* with the paraphrase 'is not surprising in the light of'. (d) As a modal particle, *auch* lacks a clearly additive meaning component. This also becomes obvious when we try to insert the stressed version of *AUCH*, which renders the sentence unacceptable:

13.   # Pia ist nicht mehr hungrig, aber Pasta hat sie **AUCH** gegessen.
      'Pia is not hungry anymore, but she also ate pasta.'

(e) Besides these semantic factors, a syntactic factor, as shown by (14), helps us to distinguish both readings. The modal particle cannot occur in preverbal position (e.g., Müller 2014). If it is placed in preverbal position, the particle can only be interpreted as the additive focus particle, which is not appropriate in the context.

14.   # Pia ist nicht mehr hungrig. **Auch** Pasta hat sie gegessen.
      'Pia is not hungry anymore. She also ate pasta.'

Thus, in (13) and (14), semantic and syntactic information help to classify the occurrence of *auch* as being the unstressed modal particle. However, although examples can be construed in which it becomes clear that the meaning of the modal particle differs from the meaning of the additive particle (see 12), we will see in the remainder of this paper that there are some cases where the two meanings are difficult to tell apart.

### 2.2. Previous Corpus Studies on Focus Particles and Focus Alternatives

Spalek and Zeldes (2017) conducted a corpus study on focus particles and the involvement of alternatives. Specifically, they investigated whether alternatives are more likely to be discussed in the subsequent context after the occurrence of the focus particle *only*. Based on experimental studies that show that focus particles such as *only* strongly activate alternative sets[4] (e.g., Gotzner et al. 2013; Gotzner et al. 2016; Spalek et al. 2014), and because of their conventional association with focus (Beaver and Clark 2008), the authors hypothesized that alternatives in naturally occurring data will be referred to more often later in the discourse if the focused element was preceded by the focus particle *only* than if it was not. The authors used the deWaC Web corpus (see Baroni et al. 2009) in order to extract different nouns (e.g., *Obst* 'fruit') that were tested in a laboratory setting before (=node words). They searched for pairs of adjacent sentences in which the first sentence contains the node word, with or without the word *nur* 'only' modifying the node's phrase. Three annotators judged where, in the context of the relevant sentence, alternatives to the node word from the first sentence could be found in the second sentence. As a more objective

test, annotators were asked to supplement the first sentence introspectively by adding 'as opposed to [ALTERNATIVE]' (e.g., 'oranges' contrast with 'apples' if it is possible to add in context: 'John likes oranges' + 'as opposed to apples'; Spalek and Zeldes 2017, p. 41). One hundred randomly selected sentence pairs were used to evaluate the annotators' ability to agree in the identification of alternatives. The inter-annotator agreement was 80%. In addition to the 100 sentence pairs annotated by three annotators, the remaining sentence pairs were divided into three random subsets for individual annotation by each of the annotators. The results indicate that the density of alternatives in the *nur* condition is significantly higher than for the sentences without *nur*. Specifically, the presence of *nur* is significantly connected to an observed increase of over 77% in the number of alternatives per sentence pair (Spalek and Zeldes 2017, p. 44). The authors conclude that whatever is viewed as an alternative to focus information in context is substantially more likely to be mentioned in the immediate subsequent context. This study thus highlights the importance of alternatives when it comes to focus particles.

Although the particle *nur*, investigated in the study by Spalek and Zeldes (2017), and the particle *auch*, investigated in the present study, are focus particles, they differ in many respects. One striking difference is the meaning of the particles in relation to their AC and the alternatives. While *nur* includes the AC (this meaning component is presupposed) and excludes alternatives (this meaning component is asserted), *auch* includes the AC (this meaning component is asserted) and also includes the alternatives (this meaning component is presupposed), as illustrated in Table 1 (see König 1991).

**Table 1.** Meaning contributions by the focus particles *also* and *only*.

|  | **Pia Also Ate Pasta.** | **Pia Only Ate Pasta.** |
| --- | --- | --- |
| Asserted (main meaning contribution) | Pia ate pasta | Nothing but pasta was eaten by Pia |
| Presupposed | Pia ate something else | Pia ate pasta |

Thus, while the reference to alternatives is the main meaning contribution in the case of the particle *nur*, it is only presupposed in the case of the particle *auch*. This leads to the question whether alternatives might be more prevalent in the case of *nur* than in the case of *auch*. Second, while *nur* always precedes its AC, there are two versions of *auch*, namely the unstressed version preceding its AC and the stressed version following its AC. Thus, it is an open question whether the involvement of alternatives differs with respect to unstressed and stressed *auch*. Furthermore, while Spalek and Zeldes looked at context sentences following the target sentence that comprised the particle *nur*, we looked at context sentences preceding the target sentence. The reason is that, compared to *nur*, *auch* usually follows its alternatives.

A further difference between the corpus study by Spalek and Zeldes and the present corpus study is that Spalek and Zeldes looked at pairs of adjacent sentences, leaving the study of larger discourse contexts as a possible point for future research. In contrast, we did not restrict our analysis to adjacent sentence pairs but looked at large discourse contexts.

We ask whether properties of (a) the alternatives and (b) their common denominators influence the choice of the speaker to use one version of the particle *auch* over the other. To address this question, we studied spoken language data, where we analyzed sentences containing the German additive particle *auch*. We (a) classified the particle as being stressed, or unstressed, (b) identified the AC of the particle, (c) searched for alternatives in the preceding context, and (d) tried to identify a common denominator, that is, a feature that is shared by the AC of the particle and the context alternatives. For cases where we were able to identify (a)–(d), we hypothesized that stressed and unstressed *auch* differ with respect to the involvement of alternatives, in that the set of alternatives should be more restricted and more explicit in the case of stressed *AUCH* than in the case of unstressed *auch*. We did not expect stressed and unstressed *auch* to differ with respect to the variability of the common denominator. Besides sentences for which we were able to classify (a)–(d), we

predicted to encounter a notable number of cases that are not classifiable. The results of the corpus study will provide insights into the role of added alternatives in the construction of utterances containing the German particle *auch*.

### 3. Materials and Methods

For the corpus study, we searched the German data base of spoken language (*Datenbank für gesprochenes Deutsch* [DGD], FOLK corpus)—a corpus management system in the area of spoken corpora, hosted by the Leibnitz Institute for the German Language (IDS). The FOLK corpus consists of private, institutional and public communication and of communication games. We selected two recordings of spoken language together with the respective transcripts: a coffee party with four speakers (FOLK_E_00201) and an oral examination with three speakers (FOLK_E_00056) (corpus search completed by the first author in May 2019). The recording of the oral examination had a duration of 33 min and 57 s, and that of the coffee party of 1 h, 37 min and 56 s. In order to have recordings with a similar duration, we only analyzed the first half of the coffee party (46 min and 8 s).

The search for occurrences of *auch* and the coding of the variables was done manually. We coded several variables, of which we will only report those that are of interest for the purpose of the present study. These variables are PARTICLE (stressed/unstressed), AC of the particle, ALTERNATIVES (explicit, reconstructable, none), NUMBER of alternatives, and DEGREE OF FREEDOM, that is, whether the common denominator of the AC and the alternative is identical, or whether they are variants of a more abstract frame of comparison (no degree of freedom/degree of freedom). We used several types of information in order to code the variables mentioned above.

### 3.1. Particle (Stressed/Unstressed) and Its AC

Based on the audio recordings we took the stress pattern into account in order to decide whether the particle *auch* was stressed or unstressed. We analyzed the whole transcripts and the entire interaction preceding the relevant occurrence of *auch* was considered to be its context. This extensive context helped us to annotate the AC of the particle. This is important, since a sentence can be ambiguous with respect to the constituent the particle associates with. In our key example 'Pia hat **auch** Pasta gegessen' (*Pia ate pasta, too*), for instance, *auch* can be stressed or unstressed. If the particle is stressed, only *Pia* can function as its AC, since stressed *AUCH* usually follows its AC (see Sudhoff 2010 for other patterns)[5]. If *auch* is unstressed, however, there are three possible ACs, as shown in (15). First, the AC can be [Pasta]. Second, the AC can be [gegessen]. And third, the AC can be [Pasta gegessen]. Intonation (indicated by capital letters) can help us to pick out [gegessen] in (15b), since this reading is signaled by a focus accent on the second syllable of *gegessen*. However, intonation is not the main cue to help us to distinguish between the possible ACs [Pasta] and [Pasta gegessen] in (15a) and (15c), since both are related to a pitch accent on the first syllable of *Pasta* (however, see Bauman et al. 2007 for tonal and articulatory marking of broad and narrow focus in German). In these examples, the presence of a context can help us to distinguish between the two ACs.

15.    a. Pia hat Pizza gegessen. Pia hat **auch** [PASta] gegessen.
       'Pia ate pizza. Pia ate pasta, too.'
       b. Pia hat Pasta gekocht. Pia hat **auch** Pasta [geGESsen].
       'Pia cooked pasta. Pia also ate pasta.'
       c. Pia hat ein Glas Rotwein getrunken. Pia hat **auch** [PASta gegessen].
       'Pia drank a glass of red wine. Pia also ate pasta.'

### 3.2. Alternatives

For detecting the added alternatives in the context, we used the test by Spalek and Zeldes (see above) and adapted it to the meaning of the focus particle *auch*: If it was possible to supplement the target sentence with 'in addition to (German: *zusätzlich zu*) [ALTERNATIVE]' (e.g., 'Pia ate pasta + in addition to [Peter]'), we sought out an alternative.

In the following, we will refer to the target sentence containing the particle and the AC and the context sentence containing the alternative as sentence pair. Note, however, that since we have taken the whole discourse context into account, target and context sentences of one sentence pair are not necessarily adjacent.

There were three categories related to the presence/absence of alternatives. If an alternative was present in the context, we classified it as explicit alternative (see 16a). If an alternative was not present but inferable from the context or from world knowledge, we classified it as reconstructable alternative (see 16b). If there was no alternative present or inferable, we classified it as no alternative (see 16c; sentences 16 and 17 are taken from the FOLK corpus). In the following examples, the AC of the particle is in square brackets.

16.   a. phraseologismen (...) werden ähm fest äh gespeichert im im k gehirn und **auch** [reproduziert].
'Idioms are fixedly stored in the brain and are also (fixedly) reproduced.'
b. (bei der modifikation) verändert sich **auch** [die bedeutung].
'In the course of modification the meaning is changing too.'
c. was vor fünfhundert jahren jemand an phraseologismus gebildet hat das äh betrifft ja oder das charakterisiert mich als modernen sprecher des deutschen doch nicht mehr also muss man da **auch** [vorsichtig] sein
'what was built as an idiom five hundred years ago no longer characterizes me as a modern speaker of German, so we have to be careful with respect to that.'

In (16a), the AC of *auch* is [reproduziert] ('reproduced') and the alternative is *fest gespeichert im Gehirn* ('fixedly stored in the brain'). In (16b), the AC of *auch* is [die Bedeutung] ('the meaning'). There is no explicit alternative present, but based on the specific knowledge that the form of an expression is changed in the course of a modification, we are able to reconstruct that the speaker might have had *die Form* ('the form') as a possible alternative in mind. Thus, it is possible to accommodate the presupposition of *auch*, that there is something else that is changed, besides the meaning of an idiom. In (16c), however, it is not possible to reconstruct an alternative, since it is not clear whether an alternative would be the negation 'not careful', or whether it would be a focus alternative to 'careful'.

In (16), all cases of *auch* are unstressed. The examples in (17) illustrate cases of explicitly mentioned alternatives (17a), reconstructable alternatives (17b) and no alternatives (17c) in the case of stressed AUCH.

17.   a. was alles dazugerechnet wird ähm das sind verben komplexe verben morphologisch und syntaktisch komplex weil sie morphologisch syntaktisch trennbar sind von ihrem erstteil und äh dieser erstteil äh sorgt für seman semantische und syntaktische veränderung an dem verb und ähm das sind aber all das schließt schon als erstes die präfixverben aus denn [die präfixverben] haben zwar **AUCH** einen erstteil der ist aber nicht trennbar.
'Complex verbs belong to this category, verbs that are morphologically and syntactically complex because they are morphologically syntactically separable from the first part. Further, this first part is the reason for semantic and syntactic changes of the verb. This excludes prefix verbs, since prefix verbs have a first part as well, however, it is not separable.'
b. aber [wenn wi mit der übertragenen bedeutung operieren] dann fallen **AUCH** wie-der ganz viele weg
'When we use the figurative meaning as an analyzing tool, we also have to exclude several other things.'
c. und ich hab in anbetracht der prüfung gesagt mein gott ich bin so aufgeregt und sie hat zu mir gesagt na aber [der] hilft dir jetz **AUCH** nich
'and thinking of the exam I said god, I am so nervous, and she said that he cannot help me either.'

In (17a), the AC of *AUCH* is [die Präfixverben] ('the prefix verbs') and the alternative is *komplexe Verben* ('complex verbs'). In (17b), the AC of *AUCH* is [wenn wir mit der

übertragenen bedeutung operieren] ('if we use the figurative meaning as an analyzing tool'). There is no explicit alternative to the figurative meaning available in the context, but one can imagine that there are other analyzing tools that can be used. Finally, in (17c), the AC of *AUCH* is [der] (he), which refers to God. There are no other persons or characters mentioned in the context and one cannot think of another relevant person or character that could be added to God in this example.

### 3.3. Number of Alternatives

In the case of explicitly mentioned alternatives, we counted the number of alternatives in the context. In (16a) and (17a), there was one alternative in the context. The utterance in (18) is an example for two alternatives in the context, with *geographischer* ('geographic') and *soziolektaler* ('sociolectal') as alternatives to *individueller* ('individual'). The example contains unstressed *auch*.

18. die aus geographischer aus soziolektaler oder aus äh ja **auch** individueller sicht ähm verschieden sein können
   'which can be different from a geographic point of view, from a sociolectal point of view and also from an individual point of view.'

Example (19) is an example for two alternatives in the context, with the speakers GB and TH as alternatives to the speaker SB. The example is an example for stressed *AUCH* (we ignore the other occurrences of *auch* here).

19. TH: da warn wir kinder aber auch noch etwas anders mama
   'we as children were also different.'
   GB: ja ihr wart auch nich so kritisch
   'yes, and you were not so critical'
   TH: und die lehrer auch
   'and neither were the teachers'
   SB: glaub [ich] **AUCH** hm hm
   'I think so too.'

Finally, there is one example for three alternatives in the context. This example contains the unstressed particle *auch*. The AC of *auch* is not realized, but an AC of the sort *gegangen* ('to leave/to go away') can be reconstructed. In the context, the alternatives *zu spät kommen* ('to be late'), *nicht vorbereitet sein* ('to not be prepared'), and *keine disziplin in die klasse bringen* ('to not discipline the class') are alternatives to 'to go away'.

20. dass da eine referentin war die jeden tach fast zu spät kam und sich nicht vorbereitet hatte und dann auch überhaupt keine disziplin in die klasse brachte und die is nachher **auch** (.) und die hat hat ihr referendariat gar nicht fertig gekricht
   'a referent who was late nearly every day, who did not prepare and who did not discipline the class and who also was . . . and who did not finish her teacher training'

There were occurrences of unstressed *auch*, where unstressed *auch* was related to a large number of alternatives, but where it was not possible to count the exact number of these alternatives. In (21), the term *ganz viele* ('many') is a vague alternative to 'collocations', which is not countable.

21. aber wenn wir mit der übertragenen bedeutung operieren dann fallen auch wieder ganz viele weg ähm weil wir ja dann au fallen **auch** [die kollokationen] weg
   'When we use the figurative meaning as an analyzing tool, we have to exclude several other things as well. We also have to exclude the collocations.'

### 3.4. Degree of Freedom

Based on the annotation of the AC and the alternative, we were able to classify whether the common denominator was repeated from the context or whether there was some degree of freedom with respect to the frame of comparison. For this comparison, we looked at the target and context sentences minus the particle, the AC and the alternatives. If the alternative-less context structure and the particle-less and AC-less target structure

were identical, we annotated that there is no DEGREE OF FREEDOM. If the two structures differed, we coded that there is a DEGREE OF FREEDOM. In example (15a), repeated in (22a), for instance, the common denominator of the target sentence is literally repeated from the context. When the alternative in the context and the particle and the AC in the target sentence are deleted, the rest of the sentence is identical. This is different in (22b). Although the meaning is similar and the alternative and the AC are the same as in (22a), the remnant sentences are not identical. Both are subsumed under a more abstract frame of comparison and we are thus dealing with some (lexical) degree of freedom regarding the words *reinschaufeln* (to shovel) and *essen* (to eat).

22. a. Pia hat ~~Pizza~~ gegessen. Pia hat **auch** [Pasta] gegessen.
    'Pia ate pizza. Pia ate pasta, too.'
    b. Pia hat ~~Pizza~~ gegessen. Pia hat sich **auch** [Pasta] reingeschaufelt.
    'Pia ate pizza. Pia also shoveled pasta into her mouth, too.'

Note that the specification whether there is some degree of freedom or not was only possible for sentences with explicitly mentioned alternatives. The following examples are cases of unstressed *auch* without (23) and with some degree of freedom (24), and cases of stressed *AUCH* without (25) and with some degree of freedom (26). The cases with an identical common denominator are mostly elliptical structures.

23. und da ham sie ja sehr viel äh notiert ähm und **auch** [einiges an literatur angegeben]
    'you wrote down a lot and you also cited a lot'
24. aus welchen wortarten sie sich speisen das sind ähm vor allen dingen also die produk-tivsten sind äh präpositionen und adverbien. (...) es können s können **auch** vereinzelt [adjektive und substantive] dazu kommen
    'which word classes they are composed of are mainly, well the most productive ones are prepositions and adverbs, but occasionally they can be complemented by adjec-tives and nouns'
25. das stimmt denk ich auch zumind also zumindest was bestimmte denotate an geht bestimmte (.) zum beispiel [die unikalen komponenten] eben **AUCH**
    'I think that this is true, at least with respect to specific meanings, for instance, with respect to the unique components'
26. also f ähm für sie palm sind ähm nur phraseologismen die nich satzwertig sind ähm tatsächlich auch phraseologismen (...) sprichwörter und geflügelte wörter fallen weg (...) genau [was] palm **AUCH** ausschließt is die tatsache dass ein phraseologismus nur aus synsemantika besteht
    'for Palm idioms are only idioms if they do not have the status of a sentence, proverbs and dictums are dropped, what is also excluded by Palm is the fact that an idiom is only comprised of function words'

In (23), *sehr viel notiert* ('write down a lot') is the alternative to *einiges an literatur angegeben* ('cite a lot') and the common denominator is *da ham sie ja x* ('you did x'). In (24), *präpositionen* ('prepositions') and *adverbien* ('adverbs') are the alternatives to *adjektive und substantive* ('adjectives and nouns'). The remnant structure is different: *das sind x* ('that are x') and *können x dazu kommen* ('can be complemented by x'). In (25), *bestimmte denotate* ('specific meanings') is the alternative to *unikale komponenten* ('unique components') and the common denominator is *das stimmt was x betrifft* ('that is true with respect to x'). In (26), the AC is *was* ('what'), which refers to *die tatsache dass ein phraseologismus nur aus synsemantika besteht* ('the fact that an idiom is only comprised of function words'). The alternative is *sprichwörter und gefügelte wörter* ('proverbs and dictums'). The remnant structure is again different (*fallen weg* ('be dropped') and *ausschließen* ('exclude')).

*3.5. Data Annotation and Exclusion of Modal Particle Uses*

The data annotation was done in two steps. In the first step, both authors coded the data independently. In the second step, both authors compared and discussed their annotations. There were three possible outcomes: (a) both annotators agreed in the first

place, (b) both annotators disagreed in the first place but agreed on one annotation after the discussion, and (c) both annotators disagreed in the first place and did not agree on one annotation. Altogether, there were 237 annotations. In 179 cases, (76%) the annotators agreed in the first place, in 34 cases (14%), they disagreed in the first place but then agreed on one annotation, and in 18 cases (7.5%) the annotators disagreed completely. Six cases (2.5%) were not classifiable. We conclude that the inter-annotator agreement is 90%, which is relatively high (see Spalek and Zeldes for an inter-annotator agreement of 80%).

For the remainder of the analysis, we excluded those cases where the annotators did not agree and those that were not classifiable (total of 24), leaving 213 cases. In one case, the particle *auch* was part of a phrase that is kind of idiomatic (*die macht ja auch was mit*, word-by-word: she makes PRT PRT something with, 'she is going through a hard time') and was therefore excluded, leaving 212 cases for the final analysis.

Furthermore, since we were interested in the focus particle readings of *auch*, we excluded all modal particle uses. It was not always trivial to distinguish the unstressed focus particle from the unstressed modal particle, since in many cases the meaning differences are subtle. Furthermore, both meanings are related, which makes it even harder to tell both meanings apart (see, for instance, Dörre and Trotzke 2019 for a meaning relation in the case of the German focus and modal particle nur 'only'). Therefore, we used the following semantic and syntactic criteria in order to distinguish the unstressed focus particle *auch* from its modal counterpart as objectively as possible.

Since modal particles do not associate with a constituent but scope over the whole proposition, they also lack alternatives. Occurrences without alternatives were further analysed. If the following additional criteria were met, the particle was counted as modal: The particle does not have an AC, it is possible to include the paraphrase 'which is not surprising in the light of'/'which is not in contrast to', the unstressed *auch* cannot by replaced by stressed *AUCH*, and the particle cannot occur (together with a potential AC) in preverbal position. Example (27) illustrates the application of these tests.

27.  (Context: The examiner tells the examinee during an oral examination that the answer the examinee gave was wrong. The examinee replies:) nee ich glaub ich hab mich da **auch** verhaspelt.
'I think I floundered (and this is the reason why I gave a wrong answer).'

In (27), *auch* is unstressed and it does not have an AC. It neither associated with *verhaspelt* ('floundered', with the possible alternative of having done something else), nor with *da* ('at that point in time', with a possible alternative of another point in time during the examination where the examinee floundered). The pronoun *ich* ('I', with the possible alternative of someone else having floundered) can also be excluded. Furthermore, it is possible to include a paraphrase of the modal particle meaning, as shown by (28).

28.  dass ich eine falsche antwort gegeben habe ist nicht überraschend in anbetracht der tatsache, dass ich mich verhaspelt habe
'that I gave a wrong answer is not surprising in light of the fact that I floundered'

The sentence becomes infelicitous when the unstressed *auch* is replaced by stressed *AUCH* (see 29), or when the particle together with its potential AC (the following constituent, which is *verhaspelt*) is placed in the prefield (see 30).

29.  #nee, verhaspelt hab ich mich da **AUCH**.
30.  #**auch** verhaspelt habe ich mich da.

The application of these tests led to the classification and exclusion of 22 modal particles. However, we want to stress that there are still cases where it is not clear whether we are dealing with the focus particle or the modal particle reading. These are cases where we have an additive meaning component on the illocutionary level, as in the case of verum focus (see Höhle 1992 for verum focus and Dimroth 2004 for verum focus in relation to auch) (indicated by capital letters, see 31).

31.  JG: ( . . . ) dann fallen auch wieder ganz viele weg ähm weil wir ja dann au fallen auch die kollokationen weg

'In this case we have to exclude several other things as well. We also have to exclude the collocations.'

AP: das stimmt das stimmt aber die kollokationen zum beispiel STEHN ja **auch** dazwischen

'That is correct, but the collocations ARE an intermediary case.'

All tests point to the modal particle reading. As a modal particle, *auch* functions to signal that the fact that collocations have to be excluded is not surprising due to the observation that they cannot be reliably classified. However, we can also detect an additive reading. As a focus particle, *auch* adds the observation that they are an intermediary case (and therefore that they have to be excluded) to the observation that they should be excluded. In other words: They should be excluded AND they have to be excluded. These cases illustrate the strong relation between modal particle readings and focus particle readings. Furthermore, they illustrate that more tests are necessary in order to distinguish between both readings.

## 4. Results

Of the 190 occurrences of the particle *auch*, the majority of 131 cases was unstressed (69%) and 59 were stressed (31%). As illustrated in Figure 1, the unstressed *auch* occurred significantly more often than stressed *AUCH* ($X^2$ = 27.28, $p < 0.001$).

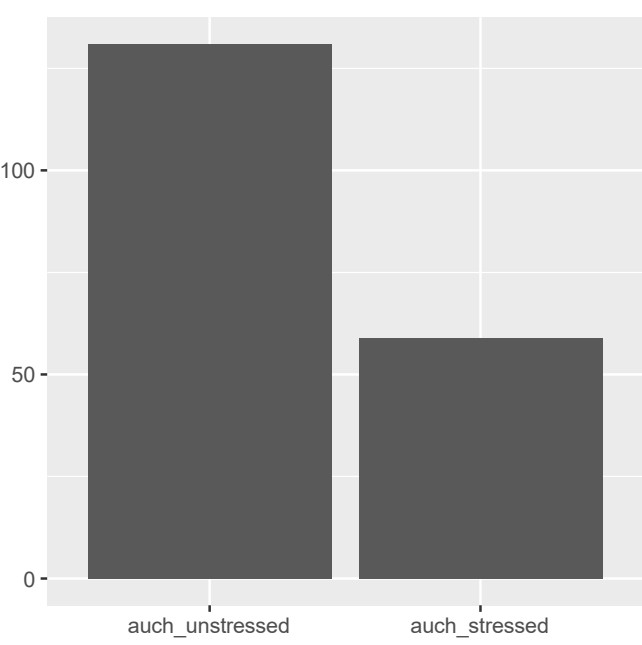

**Figure 1.** Distribution of the two versions of the particle *auch.*

In most of the cases alternatives were explicitly mentioned (58%), followed by reconstructable and no alternatives (26%, 16% respectively). However, the results show that unstressed and stressed *auch* differ with respect to the involvement of alternatives ($X^2$ = 21.35, $p < 0.001$, see Table 2 and Figure 2). While both variants of *auch* occurred most frequently with explicit alternatives (50.4% for unstressed *auch*, 75% for stressed *AUCH*), explicitly mentioned alternatives were more often present in the case of stressed *AUCH* compared to unstressed *auch*. Furthermore, while alternatives not being available were frequent in the case of unstressed *auch* (21.4%), this option was very rare for stressed *AUCH* (5%).

**Table 2.** Distribution of the alternatives over the two versions of the particle *auch*.

|  | Explicit | Reconstructable | No Alternatives |
|---|---|---|---|
| Unstressed *auch* | 66 (50.4%) | 37 (28.2%) | 28 (21.4%) |
| Stressed *AUCH* | 44 (75%) | 12 (20%) | 3 (5%) |

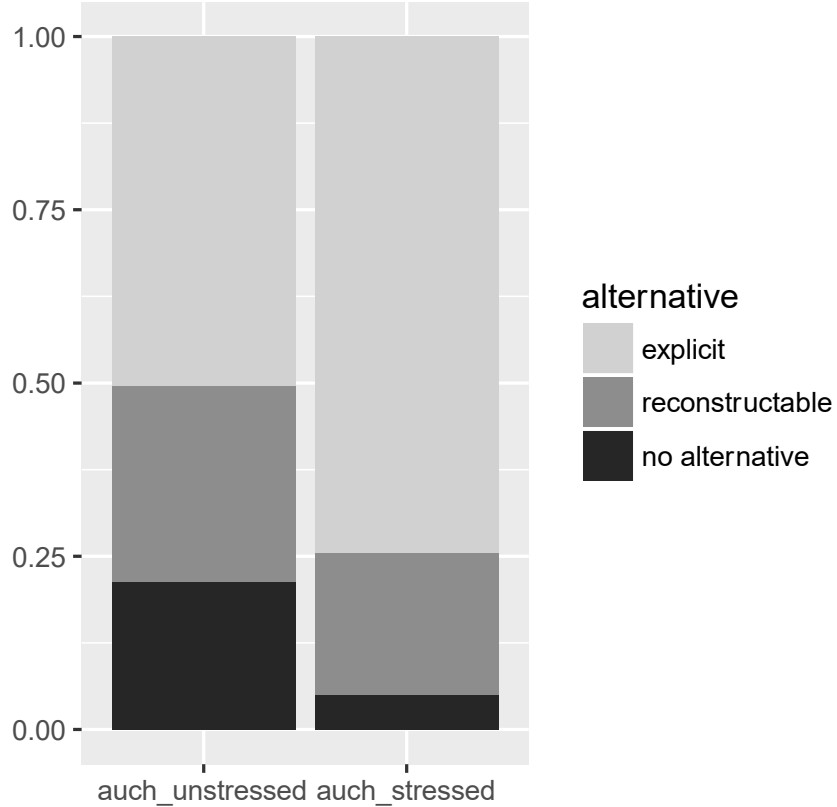

**Figure 2.** Distribution of the alternatives over the two versions of the particle *auch*.

We further analyzed the NUMBER of alternatives present in the context sentence. This was only possible for sentence pairs with explicitly mentioned alternatives. Of the 110 sentence pairs with explicitly mentioned alternatives, 100 sentence pairs involved one alternative, nine sentence pairs involved two alternatives and one sentence pair involved three alternatives (mean 1.1, SD 0.3). In most of the cases, unstressed and stressed *auch* occurred with one alternative (86% and 98%, respectively). However, while unstressed *auch* also occurred with more than one alternative (14%), this option was observed only once for stressed *AUCH* (2%, see Figure 3). In order to see whether unstressed and stressed *auch* differ with respect to the number of alternatives, we analyzed the data further by linear mixed models (R Core Team 2017, package lme4; Bates et al. 2015) with NUMBER OF ALTERNATIVES as the dependent variable, PARTICLE (unstressed *auch*/stressed *AUCH*) as fixed factor, and ITEM as random effect. There was a significant difference between unstressed and stressed *auch* ($\beta = -0.13$, SE = 0.06, t = $-2.031$, $p < 0.05$). indicating that unstressed *auch* occurs significantly more often with several alternatives than stressed *AUCH*.

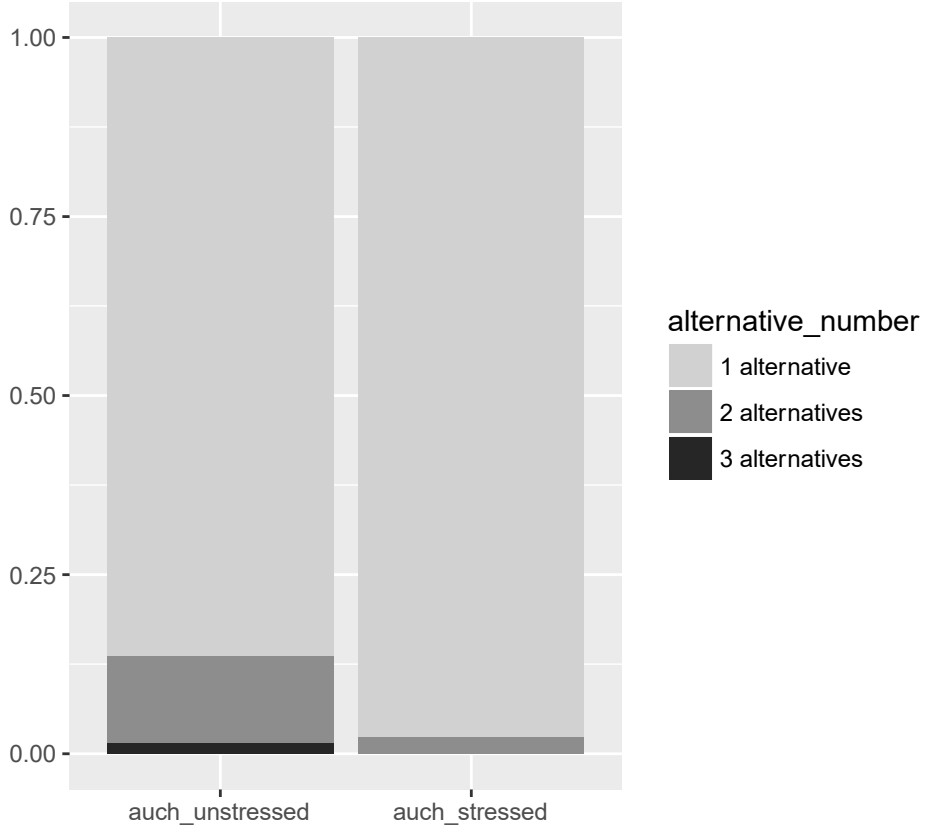

**Figure 3.** Distribution of the number of alternatives over the two versions of the particle *auch*.

For the 110 sentence pairs that involved explicitly mentioned alternatives, we looked at the common denominator of the target sentence and the context sentence. If the common denominator of the context sentence was repeated in the target sentence, we coded that there is no degree of freedom, otherwise we coded that there is a degree of freedom. The majority of the sentence pairs (64 sentence pairs) involved some degree of freedom (61%), and 41 of the sentence pairs involved the repetition of a common denominator (39%) (5 sentence pairs were not classifiable). Crucially, unstressed and stressed *auch* differ with respect to the degree of freedom (see Table 3). While unstressed *auch* equally occurs without (51%) or with (49%) a degree of freedom, stressed *AUCH* preferably occurs in sentence pairs involving some degree of freedom (79%), compared to sentence pairs having a common denominator that is repeated (21%) ($X^2$ = 13.74, $p < 0.001$, see Table 3 and Figure 4).

**Table 3.** Distribution of the degree of freedom over the two versions of the particle *auch*.

|  | No Degree of Freedom | Degree of Freedom |
|---|---|---|
| Unstressed *auch* | 32 (51%) | 31 (49%) |
| Stressed *AUCH* | 9 (21%) | 33 (79%) |

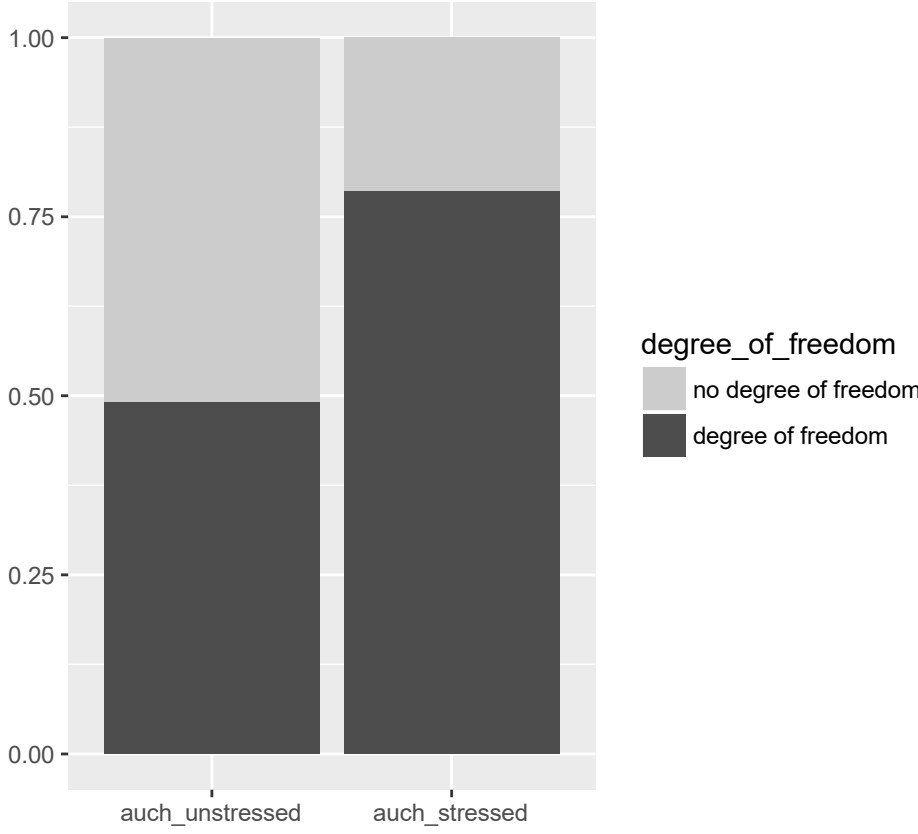

**Figure 4.** Distribution of the degree of freedom over the two versions of the particle *auch*.

## 5. Discussion

In this study, we investigated naturally occurring German conversation data containing the German additive particle *auch* and its realization as the unstressed or stressed version of the particle. By looking at the constituent, the particle associates with (the AC) and the related context alternatives, we asked whether properties of (a) the context alternatives and (b) their common denominators, that is, the feature that is shared by the AC of the particle and the context alternatives, influence the choice of the speaker to use one version of the particle *auch* over the other. This is particularly interesting since alternatives play a vital role in language comprehension and production. We conducted a corpus study on spoken language, where we analyzed 190 occurrences of the German additive particle *auch*, based on two different transcripts of spoken language with a high inter-annotator agreement of 90% (two annotators). First of all, we found that the unstressed version of *auch* is significantly more frequent than the stressed version, and that both versions differ with respect to (a) the involvement of alternatives, and (b) their common denominator. Regarding the involvement of alternatives, we found that stressed *AUCH* is mostly related to explicitly mentioned alternatives, that it often occurs with reconstructable alternatives and that it rarely occurs with no alternative. Opposed to that, unstressed *auch* is more flexible and occurs with all three constellations of alternatives. In light of the information structural differences between unstressed and stressed *auch*, this is exactly what we expected. Since the AC of stressed *AUCH* has the information structural status of a (contrastive) topic (see Krifka 1999; Dimroth 2004; Sæbø 2004), we expected that the relevant alternatives to the AC must somehow be under discussion, or, in other words, must be explicitly mentioned, or at least be reconstructable. Furthermore, stressed *AUCH*, with its utterance meaning LIKEWISE (see Reis and Rosengren 1997) marks that a similar claim is made about two distinct units of topical information. Due to the "under discussion" constraint, we expected that there are less alternatives than in the case of unstressed *auch*, where the AC has the information structural status of a focus. This was borne out by the results of the corpus

study, in that the number of explicitly mentioned alternatives was significantly smaller in the case of stressed *AUCH* than in the case of unstressed *auch*. Thus, stressed *AUCH* is related to a more restricted set of alternatives, that is explicitly mentioned, while unstressed *auch* is more flexible in this respect. Note that unstressed *auch* is highly ambiguous, since it can lead to a focus particle or a modal particle reading. Both meanings are often hard to tear apart. By using objective semantic and syntactic tests, we sorted out modal particle readings as good as possible in order to provide data that are based on focus particle readings alone. Nevertheless, future studies should go into more detail, since more objective tests are needed to distinguish between the different readings. We are convinced that future corpus linguistic investigations on particles would benefit from such tests.

Regarding the common denominator, that is, the feature that is shared by the AC of the particle and the context alternatives, we did not expect a difference between stressed and unstressed *auch.* However, the results show that there is a difference, in that stressed *AUCH* occurs with some degree of freedom in most of the cases, while there is no such preference for unstressed *auch*. Although this result is surprising and such a difference has not been observed before, the behavior of unstressed *auch* matches the results reported above regarding the alternatives, in that unstressed *auch* is again undetermined. In fact, it is at chance whether unstressed *auch* occurs with some degree of freedom or not, while it is rather determined that stressed *AUCH* occurs with a degree of freedom. Therefore, stressed *AUCH* is less strict in the sense that the remnant structure of the alternative sentence and the target sentence comprising *AUCH* does not have to be identical. An anonymous reviewer pointed out that we might be dealing with some sort of a trade-off effect: Stressed *AUCH* is stricter with respect to the alternatives, so it can be less strict with respect to the common denominator. No such relation can be observed for unstressed *auch*.

That stressed *AUCH* is chosen when there is some degree of freedom, that is, when there is a more abstract frame of comparison instead of a repetition of a common denominator, can be explained by its utterance meaning LIKEWISE. Its function is to mark two utterances as being similar, that, at least in some cases, would not have been interpreted as similar without the additive particle. In (32), for instance, which is part of a conversation of the coffee party, KH ascribes Denise B. and Felicitas Z. the same property. He puts 'people at an advanced age' on a level with 'pensioners', merely by using stressed *AUCH*. Here, the rest of the target sentence (is a pensioner) is mapped on the alternative of the context sentence (Denise B./people at an advanced age). Thus, 'people at an advanced age' and 'pensioner' are interpreted as coreferential (see Baumann and Riester 2012).

32.  (Context: TH and KH (besides GB und SB) talk about Denise B. and her medical treatment in a clinic)
     TH: ham solche leute in so_m hohen alter gar keine möglichkeit mehr in eine normale rehaklinik zu kommen
     'Those people at such an advanced age do not have the possibility anymore to be treated in a normal rehabilitation clinic'
     KH: [...] totaler unfug ist das [...] guck felicitas z. an, [die] is doch **AUCH** rentnerin [...] und [...] die sind eh alle in der bergedorf klinik in laufenburg gewesen ne orthopädische fachklinik
     'This is nonsense. Look at Felicitas Z., she is a pensioner, too. And they have been treated in the Bergedorf clinic in Laufenburg, which is an orthopedic specialist clinic.'

Another example in (33) is from the oral examination. Right at the beginning of the examination, the examinee (AP) asks whether she is allowed to use her position paper during the examination and the examiners (JG and SA) reply positively that she is allowed to use it. Furthermore, one examiner (SA) states that she herself would not be able to recall the examples of this very detailed position paper, either. However, the examinee does not claim that she is not able to recall the examples—rather, the examiner infers that the examinee is not able to recall the examples and attributes this fact to the examinee by using the particle *AUCH*.

33. AP: hab ich das recht auch mein thesenpapier zu nutzen
    'Am I allowed to use my position paper?'
    JG: natürlich
    'Of course you are.'
    SA: vor allen dingen wenn_s um beispiele geht [...] sie haben ja n sehr ausführliches thesenpapier
    'Especially with respect to your examples. You have a very detailed position paper.'
    JG: [ . . . ] sie ham ja sehr viel aufgeschrieben
    'You wrote a lot.'
    SA: ja die beispiele sind und das dürfen sie natürlich also s können [wir] **AUCH** nichso aus der lamäng
    'Yes, the examples are ... and of course you are allowed to, we are not able to do this without help, too.'

In both cases, we can clearly find the AC of the particle (Felicitas Z. in (32) and the pronoun *wir* ('we') referring to the examiners in (33)) and we find explicitly mentioned alternatives in the context (Denise B./people at an advanced age in (32) and the examinee in (33)). However, what is claimed about the alternative and the AC actually differs between the context and the target sentence, but is marked as being similar by means of stressed *AUCH*. Future studies should further examine these cases in order to investigate the exact relation between the context sentence and the sentence comprising the particle.

In our corpus study, we observed a higher inter-annotator agreement (90%) than reported for the study of Spalek and Zeldes (2017) (80%). A reason for the difference might be that we took the whole discourse context into account, which is effective, at least in the case of the particle *auch*. We observed a lot of cases where the alternatives were preceding the particle and its AC by several sentences, and we observed a lot of cases where alternatives were not present at all. Based on the semantic properties of the focus particles *auch*, which was investigated in the present study, and *nur*, which was investigated by Spalek and Zeldes, we assumed a different relation of the respective particles to alternatives (see Table 1). While the exclusion of alternatives is the main meaning contribution in the case of the particle *nur*, an additive relation to alternatives is only presupposed in the case of the particle *auch*. Thus, we expected to find some flexibility of *auch* with respect to the relation of the particle to its alternatives, which was borne out of the results of the present study. Furthermore, although studies show that an unsatisfied presupposition (i.e., an absence of an alternative and a common denominator in the context) leads to a slower processing of sentences containing the additive particle *auch* (see Schwarz 2007), it is nevertheless possible to accommodate the presupposition. Different types of presupposition triggers differ in their ease and their possibility of being accommodated, with, for instance, the definite article being easier to accommodate than the additive particle *auch* (e.g., Kripke 2009). For that reason, *auch* has been classified as being hard to accommodate. However, an accommodation is possible in the case of *auch*, especially if the context is plausible (see Singh et al. 2016), and if the presupposition makes the text more coherent (see Grubic and Wierzba 2019). The relatively high number of cases where *auch* occurs without a context alternative, or with alternatives that are not present but that can be reconstructed, suggests that it is generally possible to accommodate the presupposition of *auch* (see also example 6a). Speakers never signaled a misunderstanding in relation to the particle. The cases where *auch* is used without alternatives are cases where the interpretation is rather vague. This leads to the impression that the use of *auch* without discourse alternatives is kind of a strategy to pretend that there are alternatives, or to connect with previous parts of the utterance. The first strategy has been found in the setting of the oral examination (see 34), and the second strategy in the coffee party (see 35). Future studies should examine cases of unstressed *auch* without discourse alternatives more closely.

34. ähm es ist ja **auch** so dass (...)
    '(Besides other things I know,) it is the case that (...)'

35.  GB: so im kleinen wird das ja jetzt auch praktiziert durch den familientisch ne, da
warn
'On a small scale it is practiced by the family table'
KH: glaub das aber auch nur ne showveranstaltung. also davon halt ich überhauptnix.
das is aus meiner sicht
'I believe that this is merely a show event. I don't think much of it. From my
perspective . . . '
GB: es is aber so. ich denk **auch** an [lisa] für lisa ist das ja mal ne abwechslung
'But that's how it is. I think of Lisa, for Lisa it is a change.'

The results of our corpus study provide insights into the involvement of added
alternatives. The study reveals that alternatives strongly influence the continuation of
the discourse, in a way that the choice of stressed or unstressed *auch* depends on the
presence/absence of alternatives in the larger discourse context and on the size of the
alternative set. This is line with previous studies that emphasize the vital role of alternatives
in the course of language comprehension and production. At the same time, our study
opens the path for future research on cases with no explicit alternatives, with varying
degrees of freedom concerning the shared information between the context sentence and
the sentence hosting the particle, and on the distinction between different readings of these
very frequent and highly ambiguous elements.

## 6. Conclusions

The goal of this corpus study was to reveal to what extent the presence of alternatives
influences the continuation of the discourse, measured by the choice of the two variants
of the German particle *auch*. The results, based on the annotation of 190 natural language
occurrences of the particle *auch*, reveal an important role of context alternatives on the
continuation of the discourse, and therefore on the utterance planning of the speakers. In
most of the cases, speakers of German can choose between an unstressed and a stressed
version of *auch,* and the presence or absence of alternatives, the size of the alternative set
and the similarity between the sentence containing the particle and the context sentence
containing the alternative seem to influence the choice of the speakers.

**Author Contributions:** Conceptualization, L.R. and C.D.; methodology, formal analysis, and investi-
gation, L.R. and C.D.; writing—original draft preparation, L.R.; writing—review and editing, L.R.
and C.D. All authors have read and agreed to the published version of the manuscript.

**Funding:** This research received no external funding.

**Data Availability Statement:** The data presented in this study are available on request from the
corresponding author.

**Conflicts of Interest:** The authors declare no conflict of interest.

## Notes

[1]   Note that there is a controversy with respect to the question whether the context must provide an alternative proposition (e.g.,
Peter ate Pasta; see Beaver and Zeevat 2007; Chemla and Schlenker 2012) or whether an alternative to the focused constituent is
sufficient (e.g., Peter; see Heim 1992; Geurts and Sandt 2004). Grubic and Wierzba (2019) argue that the first account is too strong,
and for the purpose of the present study we do not assume that the alternative has to be a proposition.

[2]   One anonymous reviewer pointed out that the unacceptability seems to be partly due to the word order of the example, and that
the acceptability of the sentence increases when the word order is changed, as in '*Es ist anzunehmen, dass die Bundeskanzlerin der
Anstieg der Arbeitslosenzahlen **AUCH** beschäftigt.*' According to our intuition, however, the sentence with the unstressed version of
*auch* is still more acceptable.

[3]   The relation between the meaning of modal particles and the meaning of their counterparts (mostly focus particles and adverbs)
can be a polysemous one with a semantic relation, or a homonymous one with two different lexical entries. We assume that there
is a polysemous relation between *auch* as a focus particle and *auch* as a modal particle (for an analysis in terms of polysemy, see
Dörre and Trotzke 2019 for the German focus and modal particle nur, and Meibauer 1994 for the German modal particle and
adverb schon).

[4]   Note that besides focus particles, focus (e.g., contrastive focus) can activate a set of focus alternatives (Braun and Tagliapietra 2010).

[5]   Note that sentences containing stressed *AUCH* can also be ambiguous with respect to a possible AC when more than one potential ACs precede the particle.

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
