# Peer review of "Added Alternatives in Spoken Interaction: A Corpus Study on German Auch"

_languages, doi:10.3390/languages6040169_

Round 1

Reviewer 1 Report

This a very interesting and relevant contribution. It sheds new light on the discussion of the two variants of the focus particle auch. The paper is particularly valuable as it confronts the many existing theoretical approaches with the quirky reality of spoken interaction. Another important achievement is that the paper presents empirical evidence for Krifka’s contrastive topic hypothesis (although this could be made more explicit in the text).

The only major drawback of the paper is that in the first part of the results section (“Stressed versus unstressed additive auch” – the sections are not numbered in my version), it is not differentiated between the unstressed focus particle and modal particle uses. The numbers given for the unstressed particles include focus particles as well as modal particles, which makes a direct comparison between the stressed and unstressed focus particles impossible. For example, the frequency of explicit alternatives reported on page 8 (43% for unstressed auch, 75% for stressed AUCH) would be much more relevant if only the focus particle uses were considered. In my opinion, the comparison between the two variants of the focus particle is the most interesting aspect of the paper. My suggestion is to shorten the discussion of the general results (including modal particles) at the beginning of the results section, to continue with the differentiation between focus and modal particles, and to return to the comparison between the stressed and unstressed particles at the end of the results section, now excluding the identified modal particles.

Some minor comments and suggestions are included in the attached pdf of the article. My recommendation to the editors is to publish the paper with minor revisions.

Author Response

Dear Reviewer,

thank you so much for your comments, which we have addressed in the following way:

This a very interesting and relevant contribution. It sheds new light on the discussion of the two variants of the focus particle auch. The paper is particularly valuable as it confronts the many existing theoretical approaches with the quirky reality of spoken interaction.

Another important achievement is that the paper presents empirical evidence for Krifka’s contrastive topic hypothesis (although this could be made more explicit in the text).

We did not make it more explicit, since we did not test this hypothesis explicitly. That stressed AUCH is related to a more restricted set of alternatives is line with this hypothesis, but also with the hypothesis by Reis & Rosengren. Future studies are needed here.

The only major drawback of the paper is that in the first part of the results section (“Stressed versus unstressed additive auch” – the sections are not numbered in my version), it is not differentiated between the unstressed focus particle and modal particle uses. The numbers given for the unstressed particles include focus particles as well as modal particles, which makes a direct comparison between the stressed and unstressed focus particles impossible. For example, the frequency of explicit alternatives reported on page 8 (43% for unstressed auch, 75% for stressed AUCH) would be much more relevant if only the focus particle uses were considered. In my opinion, the comparison between the two variants of the focus particle is the most interesting aspect of the paper. My suggestion is to shorten the discussion of the general results (including modal particles) at the beginning of the results section, to continue with the differentiation between focus and modal particles, and to return to the comparison between the stressed and unstressed particles at the end of the results section, now excluding the identified modal particles.

We now state explicitly that we are only interested in focus particle readings. In the methods section we report how we excluded modal particle readings, and in the results section, we only report on focus particle uses now. We further discuss the difficulty of distinguishing both meanings and stress that future studies should look more closely into the relation between both meanings and into objective tests in order to distinguish them.

Some minor comments and suggestions are included in the attached pdf of the article.

Thanks a lot for the helpful and constructive comments and suggestions. We tried to include them as good as possible.

Reviewer 2 Report

There are some problems both concerning the presentation and the content.

1) Relation between the research questions and the results: The paper repeatedly mentions that its aim is "to reconstruct criteria by which speakers construe context expressions as alternative(s)". If I understand this goal correctly, I'm not certain that this is what the paper achieves. Perhaps better research questions would be whether there are differences between stressed and unstressed additive auch with respect to whether explicit contextual alternatives are required, and in the nature of their overt context alternative?
I think a lot of the paper can remain the same, including the hypothesis: "the set of alternatives should be more restricted and more explicit in the case of stressed AUCH than in the case of unstressed auch. We did not expect stressed and unstressed auch to differ with respect to the flexibility of the common denominator."

I think that the fact that a lot of examples of additive auch (both kinds) were found where the alternatives are not explicit but reconstructable or even missing entirely is interesting and should be highlighted more. The same is true for the "degree of freedom" examples. I'll come back to this point below.

2) A more minor issue is that the paper also presents results concerning "modal" uses of the particle but this is not mentioned as one of the research questions. I was also a bit confused by the results concerning "modal" uses. I find it convincing that other criteria (e.g. prosody) are used to distinguish stressed and unstressed additive "auch". So I was puzzled that the modal use was basically identified via "not being additive" - if you look at the 44 or so occurrences of unstressed "auch" without an additive reading, do they all have the relevant properties of modal particles that you discuss in (9) or are there instances that are not easy to classify? For me, it would be clearer and nicer to first check (using independent criteria) which instances of "auch" are classified as modal by the annotators (i.e. to make a four-way distinction between stressed additive auch, unstressed additive auch, modal auch and other/non-classifiable), and then to check the properties of the context. This would also lead to clearer results for unstressed additive "auch" (since when the cases of "modal" auch are taken out of the set, the percentages of cases with explicit vs. reconstructable vs. no alternatives will change, too).

I think there are two ways to amend this (my preferred way would be (i)): (i) clarify the empirical part on "modal" auch and make this part more prominent by adding a research question on modal auch, (ii) shorten the part on "modal" auch: just include a category like "other (non-additive) uses of auch", and say that the 44 instances belong to this category.

3) an even more minor comment is that I don't quite understand why the position of particle and AC (preverbal / postverbal) was annotated - this is one of the cases where no difference between stressed and unstressed auch is expected, right? I would clarify (without lengthening too much) why exactly these two positions are relevant. Does it have something to do with topicality?

The same holds for scalar meanings: Since - I believe - scalar readings of "auch" are additive too, I don't understand why this was investigated. Since no such examples were found, perhaps place in a footnote?

4) It would be important and interesting to see more corpus examples. At the moment, the readers are merely presented with the numbers and very few examples. I would have liked to see examples of all classifications mentioned (e.g. unstressed auch and stressed AUCH with explicit, reconstructable and no alternatives; unstressed auch and stressed AUCH with one vs. two alternatives and the example with three alternatives; unstressed auch and stressed AUCH with identical and non-identical common denominator).

I have never worked with corpora and don't know what legal issues might prevent this, but ideally some possibility for readers to access the complete set of categorized data should be provided (e.g. as supplementary files, as a link to a website listing the relevant occurrences in their respective category, or perhaps only as a list of references to the examples for those readers who have access to the corpora).

5) About the notion of contextual alternatives: perhaps mention explicitly when introducing contextual alternatives that this is not a proposition - in the literature often the contextually provided alternative is often seen to be a full proposition.

Perhaps something to clarify with respect to the "flexibility of contextual alternatives" would be: is the main claim that alternatives need not be present in the immediate (linguistic or non-linguistic) context? In that case, it may make sense to cite an example where no context is necessary (usually in contexts where there is only one viable other alternative than the AC), e.g. (Beaver & Zeevat 2004: "(Hey) X have feelings too(!) where X can be e.g. men, fish or we").

On the other hand, if the main claim is that sometimes there is something in the context which can be construed as a possible antecedent but doesn't make the alternatives explicit, a simpler example such as the following might suffice (since alternatives are assumed to be of the same semantic type as Paul, i.e. type e, the context does not explicitly list any such alternatives):

Many people ate pasta. Paul ate pasta, too.

In any case, since this is a claim about alternatives to the AC I would try to keep the "common denominator" part of the utterance the same in order to avoid confusion. I would also advise against ex. (4) because it is not entirely clear, in my opinion, that this is in fact narrow focus.

Perhaps you could come back to this in the discussion of your results: are there systematic differences between examples where the alternatives are not explicit?

6) On the notion of the "common denominator": If you look at the literature on Givenness (e.g. Schwarzschild 1999, Baumann and Riester 2012), it is usually assumed that entailment plays a role. For example, one should be able to replace a referring expression with one referring to the same individual (e.g. Fido by "that little rascal"):

(i) Fido destroyed my shoes. That little rascal destroyed [the couch]F, too.

Likewise, a non-referring expression in the antecedent should be replacable by a synonym or hypernym in the current utterance (e.g. apple by fruit). I believe however that there may be more flexibility if the expression can be subsumed under the same QUD background (e.g. the co-hyponyms in (ii) under "parents"):

(ii) Who of the kids brought their parents?
Andy brought his mother, and [Berta]_F brought her father, too.

It would be interesting to hear more, in the introduction/literature overview as well as in the discussion, about the flexibility of the common denominator: how flexible is it, what is possible?

7) Concerning the presentation, I would include more subchapters, especially for the introduction: I would basically provide a longer version of the abstract as a kind of introduction (including a brief discussion of the corpus study and its results), then a chapter called "literature overview" or similar which discusses previous answers to the research questions and motivates the different categories which were annotated. I would list and number some possible hypotheses for the corpus study. The sections

8) There is some literature that might be worth looking at/citing: 
p.1, ex. (2) - it might be relevant to cite Heim 1992 ("My parents think I am also in bed") for this (see also Tonhauser et al. 2013, Geurts and van der Sandt 2004).
p.1 cite some literature for the flexibility of the common denominator (perhaps check Chemla and Schlenker 2012?)
p.2 not sure whether there might be syntactic reasons why (6b) is odd?
p.3 cite sb for the observation that "auch" can have a scalar reading - I can't think of a reference on the spot, perhaps König 1991?

Author Response

Dear Reviewer,

thank you so much for your comments, which we have addressed in the following way:

First of all, I don't think that the paper is ready for publication as it is, there are some problems both concerning the presentation and the content.

1) Relation between the research questions and the results: The paper repeatedly mentions that its aim is "to reconstruct criteria by which speakers construe context expressions as alternative(s)". If I understand this goal correctly, I'm not certain that this is what the paper achieves. Perhaps better research questions would be whether there are differences between stressed and unstressed additive auch with respect to whether explicit contextual alternatives are required, and in the nature of their overt context alternative?

We now changed the description of the goal of the study and the research question into “It is the aim of the present study to investigate whether number and explicitness of discourse alternatives influence the construction of utterances containing the German particle auch” and “We ask, whether there are differences between stressed and unstressed additive auch with respect to whether explicit contextual alternatives are required.”.

I think a lot of the paper can remain the same, including the hypothesis: "the set of alternatives should be more restricted and more explicit in the case of stressed AUCH than in the case of unstressed auch. We did not expect stressed and unstressed auch to differ with respect to the flexibility of the common denominator."

I think that the fact that a lot of examples of additive auch (both kinds) were found where the alternatives are not explicit but reconstructable or even missing entirely is interesting and should be highlighted more. The same is true for the "degree of freedom" examples. I'll come back to this point below.

We now included more examples.

2) A more minor issue is that the paper also presents results concerning "modal" uses of the particle but this is not mentioned as one of the research questions. I was also a bit confused by the results concerning "modal" uses. I find it convincing that other criteria (e.g. prosody) are used to distinguish stressed and unstressed additive "auch". So I was puzzled that the modal use was basically identified via "not being additive" –

“not being additive” was not our only criterion for the identification of modal particles. A second criterion was that the particle cannot be placed in the prefield. Especially the prefield test is an objective way to classify modal particles. In the current version we added a further test, namely the possibility to insert a typical paraphrase of the modal particle reading (`which is not surprising in light of the fact that’).

if you look at the 44 or so occurrences of unstressed "auch" without an additive reading, do they all have the relevant properties of modal particles that you discuss in (9) or are there instances that are not easy to classify? For me, it would be clearer and nicer to first check (using independent criteria) which instances of "auch" are classified as modal by the annotators (i.e. to make a four-way distinction between stressed additive auch, unstressed additive auch, modal auch and other/non-classifiable), and then to check the properties of the context. This would also lead to clearer results for unstressed additive "auch" (since when the cases of "modal" auch are taken out of the set, the percentages of cases with explicit vs. reconstructable vs. no alternatives will change, too).

I think there are two ways to amend this (my preferred way would be (i)): (i) clarify the empirical part on "modal" auch and make this part more prominent by adding a research question on modal auch, (ii) shorten the part on "modal" auch: just include a category like "other (non-additive) uses of auch", and say that the 44 instances belong to this category.

We now state explicitly that we are mainly interested in focus particle readings and that we excluded modal particle readings. However, in order to distinguish both readings, information on modal particles are necessary. For that reason, we lengthened the part on modal particles in the introduction and in the methods section. We describe in more detail, how modal uses were classified and we offer more examples for this. In the results section, we only report on focus particle uses now.

3) an even more minor comment is that I don't quite understand why the position of particle and AC (preverbal / postverbal) was annotated - this is one of the cases where no difference between stressed and unstressed auch is expected, right? I would clarify (without lengthening too much) why exactly these two positions are relevant. Does it have something to do with topicality?

Yes, it is related to topicality. It is assumed that the preverbal position is the contrastive topic position, and that it is possible to accommodate the presupposition of auch if auch is placed in this position, even if it is not contextually introduced (see Gotzner and Spalek 2014). However, since there were only very few cases, we excluded this part now.

The same holds for scalar meanings: Since - I believe - scalar readings of "auch" are additive too, I don't understand why this was investigated. Since no such examples were found, perhaps place in a footnote?

This is also a good point – we now excluded this criterium.

4) It would be important and interesting to see more corpus examples. At the moment, the readers are merely presented with the numbers and very few examples. I would have liked to see examples of all classifications mentioned (e.g. unstressed auch and stressed AUCH with explicit, reconstructable and no alternatives; unstressed auch and stressed AUCH with one vs. two alternatives and the example with three alternatives; unstressed auch and stressed AUCH with identical and non-identical common denominator).

We now included examples for each case mentioned above.

I have never worked with corpora and don't know what legal issues might prevent this, but ideally some possibility for readers to access the complete set of categorized data should be provided (e.g. as supplementary files, as a link to a website listing the relevant occurrences in their respective category, or perhaps only as a list of references to the examples for those readers who have access to the corpora).

We now included the exact number of the transcripts. After registering at the DGD, the FOLK corpus is freely accessible.

5) About the notion of contextual alternatives: perhaps mention explicitly when introducing contextual alternatives that this is not a proposition - in the literature often the contextually provided alternative is often seen to be a full proposition.

We now included this.

Perhaps something to clarify with respect to the "flexibility of contextual alternatives" would be: is the main claim that alternatives need not be present in the immediate (linguistic or non-linguistic) context? In that case, it may make sense to cite an example where no context is necessary (usually in contexts where there is only one viable other alternative than the AC), e.g. (Beaver & Zeevat 2004: "(Hey) X have feelings too(!) where X can be e.g. men, fish or we").

Thank you for pointing out this example, which we now included.

On the other hand, if the main claim is that sometimes there is something in the context which can be construed as a possible antecedent but doesn't make the alternatives explicit, a simpler example such as the following might suffice (since alternatives are assumed to be of the same semantic type as Paul, i.e. type e, the context does not explicitly list any such alternatives):

Many people ate pasta. Paul ate pasta, too.

In any case, since this is a claim about alternatives to the AC I would try to keep the "common denominator" part of the utterance the same in order to avoid confusion.

I would also advise against ex. (4) because it is not entirely clear, in my opinion, that this is in fact narrow focus. Perhaps you could come back to this in the discussion of your results: are there systematic differences between examples where the alternatives are not explicit?

We replaced example 4 by another example and we go into more detail with respect to the cases without explicit alternatives in the discussion section.

6) On the notion of the "common denominator": If you look at the literature on Givenness (e.g. Schwarzschild 1999, Baumann and Riester 2012), it is usually assumed that entailment plays a role. For example, one should be able to replace a referring expression with one referring to the same individual (e.g. Fido by "that little rascal"):

(i) Fido destroyed my shoes. That little rascal destroyed [the couch]F, too.

Likewise, a non-referring expression in the antecedent should be replacable by a synonym or hypernym in the current utterance (e.g. apple by fruit). I believe however that there may be more flexibility if the expression can be subsumed under the same QUD background (e.g. the co-hyponyms in (ii) under "parents"):

(ii) Who of the kids brought their parents?
Andy brought his mother, and [Berta]_F brought her father, too.

It would be interesting to hear more, in the introduction/literature overview as well as in the discussion, about the flexibility of the common denominator: how flexible is it, what is possible?

Thank you for this recommendation. We now included more details on the flexibility of the common denominator. At the same time, we leave it to future research to further annotate the data with respect to the flexibility of the common denominator and to the exact relation between the context sentence and the sentence hosting the particle.

7) Concerning the presentation, I would include more subchapters, especially for the introduction: I would basically provide a longer version of the abstract as a kind of introduction (including a brief discussion of the corpus study and its results), then a chapter called "literature overview" or similar which discusses previous answers to the research questions and motivates the different categories which were annotated. I would list and number some possible hypotheses for the corpus study. The sections

We now included subchapters and a introduction.

8) There is some literature that might be worth looking at/citing: 
p.1, ex. (2) - it might be relevant to cite Heim 1992 ("My parents think I am also in bed") for this (see also Tonhauser et al. 2013, Geurts and van der Sandt 2004).
p.1 cite some literature for the flexibility of the common denominator (perhaps check Chemla and Schlenker 2012?)
p.2 not sure whether there might be syntactic reasons why (6b) is odd?
p.3 cite sb for the observation that "auch" can have a scalar reading - I can't think of a reference on the spot, perhaps König 1991?

Thank you for these recommendations which we included now.

Round 2

Reviewer 2 Report

I think that the paper is much improved, and that my comments have mostly been addressed. Here are some final remarks:

p.2: Schwarzschild 1999 not cited with the intention that I had for proposing this reference, but perhaps this should be left to the authors.
p.3: Beaver and Zeevat: please mention explicitly here what X may be ("men, fish or we") since Y is explicitly mentioned ("women, people, you").
p.5 and p.17: "tear apart" -> "tell apart"

p.5-7, §2.2:  I'm not sure anymore whether I mentioned this in my first review or forgot, but don't quite understand why a corpus study on a different phenomenon and with different methods and aims is reviewed in such length. My recommendation would be to just shorten this section so that only the parts most relevant to the current paper remain, possibly even to delete it.

p. 7ff: I would recommend more subsections in section 3. I would recommend to first formulate and number the hypotheses gained from the preceding section(s). If this is done, it would be good to pick this up again in the Discussion chapter - which hypotheses were confirmed?

In the subsection (§3.1) "Materials and Methods" it would be great to add a sentence on the DGD - what kind of corpus is it? e.g. when collected, where, by whom?
In addition, it would be good to provide more information on how the "auch"-sentences were retrieved (partly already provided in §2), and what software was used.

A possible further subsection could start on p.11 where there is a change in topic from a characterization of the variables being annotated to the annotation procedure. 

Author Response

Dear Reviewer,

thank you so much for your comments!

p.3: Beaver and Zeevat: please mention explicitly here what X may be ("men, fish or we") since Y is explicitly mentioned ("women, people, you"). --> we included that

p.5 and p.17: "tear apart" -> "tell apart"  --> we changed that

p.5-7, §2.2:  I'm not sure anymore whether I mentioned this in my first review or forgot, but don't quite understand why a corpus study on a different phenomenon and with different methods and aims is reviewed in such length. My recommendation would be to just shorten this section so that only the parts most relevant to the current paper remain, possibly even to delete it.

This is a key publication with respect to focus particles, focus alternatives, and corpus studies. Since we also work on focus particles and focus alternatives, this study is highly relevant and we reviewed this study in such length. Additionally, we based parts of our methodology on this study (e.g., the test that helps to detect focus alternatives) and the results are relevant for the purpose of our study (e.g., the inter-annotator agreement; the increase of the number of focus alternatives when the focus particle nur is present). However, we see your point that the aim of the study is a different one, and we shortened this section.

p. 7ff: I would recommend more subsections in section 3. I would recommend to first formulate and number the hypotheses gained from the preceding section(s). If this is done, it would be good to pick this up again in the Discussion chapter - which hypotheses were confirmed?

We now included subsections.

We already present our hypothesis at the end of section 2. Therefore, we did not include them in section 3 again.

In the subsection (§3.1) "Materials and Methods" it would be great to add a sentence on the DGD - what kind of corpus is it? e.g. when collected, where, by whom?

We now included these points.

In addition, it would be good to provide more information on how the "auch"-sentences were retrieved (partly already provided in §2), and what software was used.

We now included the information that the search for occurrences of auch was done manually. We did not use any specific software.

A possible further subsection could start on p.11 where there is a change in topic from a characterization of the variables being annotated to the annotation procedure. 

Besides other subsections, we included a subsection there.
